

# Geographic origin and timing of colonization of the Pacific Coast of North America by the rocky shore gastropod *Littorina sitkana*

Peter B. Marko[1] and Nadezhda I. Zaslavskaya[2]

[1] School of Life Sciences, University of Hawaiʻi at Mānoa, Honolulu, Hawaiʻi, United States of America
[2] National Scientific Center of Marine Biology, Far Eastern Branch, Russian Academy of Sciences, Vladivostok, Russian Federation

## ABSTRACT

The demographic history of a species can have a lasting impact on its contemporary population genetic structure. Northeastern Pacific (NEP) populations of the rocky shore gastropod *Littorina sitkana* have very little mitochondrial DNA (mtDNA) sequence diversity and show no significant population structure despite lacking dispersive planktonic larvae. A contrasting pattern of high mtDNA diversity in the northwestern Pacific (NWP) suggests that *L. sitkana* may have recently colonized the NEP from the NWP via stepping-stone colonization through the Aleutian-Commander Archipelago (ACA) following the end of the last glacial 20,000 years ago. Here, we use multi-locus sequence data to test that hypothesis using a combination of descriptive statistics and population divergence modeling aimed at resolving the timing and the geographic origin of NEP populations. Our results show that NEP populations share a common ancestor with a population of *L. sitkana* on the Kamchatka Peninsula ~46,900 years ago and that NEP populations diverged from each other ~21,400 years ago. A more recent population divergence between Kamchatka and NEP populations, than between Kamchatka and other populations in the NWP, suggests that the ACA was the most probable dispersal route. Taking into account the confidence intervals for the estimates, we conservatively estimate that *L. sitkana* arrived in the NEP between 107,400 and 4,100 years ago, a range of dates that is compatible with post-glacial colonization of the NEP. Unlike other congeners that are relatively abundant in the Pleistocene fossil record of the NEP, only one report of *L. sitkana* exists from the NEP fossil record. Although broadly consistent with the molecular data, the biogeographic significance of these fossils is difficult to evaluate, as the shells cannot be distinguished from the closely-related congener *L. subrotundata*.

## INTRODUCTION

Resolving the contemporary and historical factors affecting patterns of spatial genetic variation is a fundamental goal of population genetics and phylogeography (*Slatkin, 1985*; *Avise, 2004*). For neutral genetic loci, population genetic structure is expected to reflect the

Corresponding author
Peter B. Marko, pmarko@hawaii.edu

combined effects of mutation, genetic drift, and gene flow among populations. Across stable landscapes and seascapes, these evolutionary forces may result in incremental evolutionary divergence among populations, eventually reaching a dynamic evolutionary equilibrium. Large changes to the environment, particularly those that result in major changes in the abundance and distribution of species, may leave long-lasting impacts on spatial patterns of genetic diversity (*Whitlock & McCauley, 1999*; *Neigel, 2002*). Genetic data are therefore of great use in the field of biogeography, as a tool to understand the demographic history of species.

Inferences about the historic biogeography of plants and animals have relied heavily on mitochondrial DNA (mtDNA) sequence data. In addition to being relatively easy to amplify and sequence, the haploid structure and maternal mode of inheritance in most animals results in an effective population size that is one-quarter that of chromosomes in the nuclear genome (*Avise et al., 1987*). Combined with a relatively high mutation rate, these unusual properties lead to rapid evolutionary divergence between populations, making mtDNA an indispensable and inexpensive tool for detecting population structure, even as genomic data become more common in studies of population genetics and biogeography (*Bowen et al., 2014*; *Hung, Drovetski & Zink, 2016*; *Hurst & Jiggins, 2005*).

One of the major contributions of mtDNA phylogeography has been frequent characterization of low genetic diversity at high latitudes among plants and animals, a pattern widely interpreted as reflecting post-glacial colonization from low-latitude refugia (e.g., *Hewitt, 1999*; *Hewitt, 2004*; *Hofreiter & Stewart, 2009*; *Maggs et al., 2008*; *Marko et al., 2010*; *Jenkins, Castilho & Stevens, 2018*). Although selective sweeps can create the same pattern (e.g., *Wares, 2009*; *Ilves et al., 2010*), decreased mtDNA diversity at high latitudes provides a logical and consistent compliment to evidence from the fossil record that shows many large changes in the geographic distributions of species in response to glacial-interglacial climate change during the Pleistocene (*Valentine, 1989*; *Valentine & Jablonski, 1993*; *Roy et al., 1996*).

In the northeastern Pacific (NEP), mtDNA data from many (but not all) marine species are consistent with the idea that many nearshore benthic species likely retreated to southern refugia during the last glacial maximum (LGM), only recently returning to higher latitudes during the present warm interglacial (e.g., *Hellberg, 1994*; *Marko, 1998*; *Edmands, 2001*; *Hickerson & Cunningham, 2005*; *Marko et al., 2010*). Among NEP species with spatial patterns of genetic diversity that are consistent with recent range expansions, several have geographic ranges extending to the northwestern Pacific (NWP) (e.g., *Vermeij, Palmer & Lindberg, 1990*), leaving open the question that populations in Asia may have served as a glacial refugium for NEP populations of amphi-Pacific taxa (*Vermeij, 1989*). For one such species, the Sitka Periwinkle *Littorina sitkana* (Philippi 1846), population genetic sub-structuring is nearly non-existent in the NEP (*Kyle & Boulding, 2000*; *Sokolova & Boulding, 2004*; *Lee & Boulding, 2009*; *Marko et al., 2010*; *Botta et al., 2014*), but much greater over relatively small spatial scales in the NWP (*Nohara, 1999*; *Zaslavskaya & Pudovkin, 2005*; *Azuma et al., 2017*). Based on these contrasting patterns of diversity and the observation that the most common mtDNA haplotype in the NEP is identical to one

found in the NWP, *Azuma et al. (2017)* proposed that NEP populations of *L. sitkana* must have been derived from the NWP following the end of the last glacial.

Here, we use multi-locus sequence data set to test this hypothesis and to estimate the divergence time of NEP and NWP populations of *L. sitkana*. Although the smaller effective population size and the greater sensitivity to genetic drift make mtDNA especially useful for detecting range expansions, this aspect of mtDNA evolution, which makes it a "leading indicator" of evolutionary divergence (*Zink & Barrowclough, 2008*), comes at the cost of rapid loss of polymorphisms within populations and loss of information about demographic parameters such as ancestral population size, gene flow, and divergence times with other populations. Even though the growing use of genomic datasets provides access to thousands of single nucleotide polymorphisms, a handful of additional nuclear sequence loci, from which gene trees can be constructed, may provide sufficiently robust estimates of population genetic parameters that can be used to distinguish between alternative biogeographic histories (*Marko & Hart, 2011*). To that end, we have combined previously collected sequences with new multi-locus data from populations of *L. sitkana* in the NEP and NWP and have analyzed the data with a combination of descriptive population genetic statistics, approximate Bayesian computation modeling of biogeographic histories, and coalescent-based modeling of population divergences.

## MATERIALS & METHODS

### Samples and DNA extraction

Specimens of *Littorina sitkana* were collected from 14 locations in the summers of 2002 and 2008, seven in the northwestern Pacific (NWP) and seven in the northeastern Pacific (NEP) (Table 1). Collections in Alaska were obtained with permission from the Alaska Department of Fish and Game. In Japan, collections were approved in person at local offices of the Hokkaido Federation of Fisheries Cooperative Associations in Erimo, Nemuro, and Utoro. Collections of marine invertebrates in Russia required no permits. To ensure tissue preservation, each shell was cracked with a small hammer before being placed in ethanol. Genomic DNA was extracted from each individual using a standard CTAB protocol (*Marko, Rogers-Bennett & Dennis, 2007*), re-suspended in water, and stored at $-20\,°C$.

### PCR and sequencing

Using previously published primers (*Folmer et al., 1994*; *Jarman, Ward & Elliott, 2002*; *Sokolova & Boulding, 2004*) and methods (*Marko et al., 2010*), we amplified and sequenced four loci, including 433 bp of mitochondrial cytochrome b (*CYTB*) and introns from three nuclear loci: 881 bp of adenosine triphosphate synthase subunit alpha (*ATPSα*), 612 bp of adenosine triphosphate synthase subunit beta (*ATPSβ*), and 597 bp of aminopeptidase (*APN54*). *CYTB* sequences were collected from new NWP samples (VOS, ERI, NEM, UTO, KHO, STA, PET) but previously published *CYTB* sequences were electronically-retrieved for samples from NEP populations (SJI, REN, RUP, CAM, COR, KOD, JUN; GenBank: GQ902686–GQ902751). Mitochondrial DNA (mtDNA) was directly sequenced with both primers, assembled into contigs and edited with Sequencher 4.8 (GeneCodes Corp.), and aligned with other consensus sequences.

**Table 1  Sample sites.** Localities and sample sizes for phylogeographic analysis of *Littorina sitkana*.

| Locality | N | Latitude | Longitude | Date |
|---|---|---|---|---|
| Vostok Bay (VOS) | 17 | 42.87N | 132.74E | June, 2008 |
| Erimo-misaki (ERI) | 11 | 41.93N | 143.23E | June, 2008 |
| Nemuro-misaki (NEM) | 14 | 43.38N | 145.78E | June, 2008 |
| Utoro-misaki (UTO) | 9 | 44.06N | 144.98E | June, 2008 |
| Kholmsk (KHO) | 16 | 47.05N | 142.05E | July, 2008 |
| Starodubskoye (STA) | 8 | 47.40N | 142.81E | June, 2008 |
| Petropavlovsk-Kamchatsky (PET) | 16 | 52.91N | 158.75E | July, 2008 |
| Kodiak Island (KOD) | 9 | 57.83N | 152.41W | May, 2008 |
| Cordova (COR) | 12 | 60.59N | 145.75W | May, 2002 |
| Juneau (JUN) | 8 | 58.30N | 134.43W | May, 2002 |
| Prince Rupert (RUP) | 14 | 54.31N | 130.31W | May, 2002 |
| Campbell River (CAM) | 11 | 50.02N | 125.18W | May, 2002 |
| Port Renfrew (REN) | 6 | 48.55N | 124.44W | May, 2002 |
| San Juan Island (SJI) | 6 | 48.55N | 123.01W | May, 2002 |

For the nuclear DNA (nDNA), we cloned, plated, and sequenced PCR products from a subset of four localities in the west (ERI, KHO, STA, and PET) and three in the east (KOD, COR, JUN) for use in coalescent analyses and demographic modeling. PCR products were inserted into a vector with the pGEM Easy Vector System; Electromax DH10B *E. coli* cells (Promega) were transformed with the vector by electroporation. Sequences were edited and aligned with Sequencher and then aligned with ClustalX v1.83.1 (*Chenna et al., 2003*). For each individual, we selected six colonies (clones) from each plate for sequencing; if two clones from the same plate yielded different sequences, we considered that individual to be a heterozygote; individuals were scored as homozygous if all six colonies contained the same sequence. New mtDNA and nDNA data were deposited in GenBank (Accession numbers: MN120839–MN120881).

## Genetic diversity and population structure

We used Arlequin version 3.5.1.2 (*Excoffier, Laval & Schneider, 2005*) and DNAsp version 5.10.01 (*Librado & Rozas, 2009*) to calculate nucleotide ($\pi$) and haplotype ($h$) diversities, *Tajima*'s (*1989*) $D$, *Fu*'s (*1997*) $F_s$, and *Ramos-Onsins & Rozas (2002)* $R_2$. Significance of the descriptive statistics was evaluated with 10,000 permutations (Arlequin) or coalescent simulations (DNAsp) of the data. We then used an analysis of molecular variation (AMOVA) to estimate $\Phi_{ST}$ between all pairs of samples for all four loci. The best-fitting substitution model for each locus was then chosen with ModelTest version 3.8 (*Posada & Crandall, 1998*; *Posada, 2006*) using the Akaike Information Criterion. The most similar model available in Arlequin was used for the AMOVA.

## Haplotype networks

Because gene trees and coalescent population genetic analyses are usually constructed with the assumption of no intra-locus recombination, we tested the nDNA sequences for evidence of recombination using the pairwise homoplasy index ($\Phi_w$) statistic (*Bruen,*

*Philippe & Bryant, 2006*) in Splitstree 4.10 (*Hudson & Bryant, 2006*). *ATPSα* ($P < 0.035$) and *APN54* ($P < 0.01$) showed significant evidence of recombination whereas *ATPSβ* did not ($P = 0.653$). We then used an implementation of the four-gamete test (*Hudson & Kaplan, 1985*) in IMgc (*Woerner, Cox & Hammer, 2007*) to exclude putative recombinant nucleotide sites in *ATPSα* and *APN54*. After removal of recombinant sites, re-testing with the $\Phi_w$ statistic showed no evidence of recombination at either locus ($P = 1.0$). We then used PopART to construct unrooted minimum spanning networks for each locus (*Bandelt, Forster & Röhl, 1999*).

## Pairwise population divergence modeling

We used IMa2 (*Hey & Nielsen, 2004*) to estimate gene flow, population divergence times, and effective population sizes for all pairwise combinations of samples for which we collected multi-locus data. Pairwise isolation-with-migration (IM) models may be a poor fit to the data (because they ignore migration from other populations), but they have the advantage of greater power given the relatively smaller number of model parameters (compared to multi-population models of divergence). Given that *L. sitkana* does not have planktonic larvae and our samples are separated by hundreds to thousands of km, we used exponential distributions (with a mean of $0.1/\mu$) for the migration priors. Exponential priors are conservative with respect to inferring gene flow (a greatest prior probability at *m* = 0) but they do not discount the likelihood of gene flow given they lack an explicit upper bound.

IMa2 was run on the University of Hawaii High Performance Computing Cluster in 72 h intervals (the maximum length of jobs permitted on the cluster) in which the run was repeatedly restarted by loading the state space from the previous run (i.e., as burn-in, without reloading the sampled values from the previous run). After each 72-hour interval, convergence was assessed by comparing the parameter estimates and trend plots between the first and second half of the run. When a run had converged, we then restarted the run as before but with the sampled values from each previous run reloaded until 100,000 genealogies had been sampled. Convergence was also assessed by repeating each pairwise analysis three times with different random number seeds. We converted estimates of genetic diversity ($\Theta$) and divergence times (*t*) into demographic quantities (i.e., individuals and years, respectively) using a divergence rate of 1% per million years. This rate was chosen based on divergence rates inferred for protein-coding mtDNA genes in fossil calibrated phylogenies of gastropods (*Marko, Moran & Zaslavskaya, 2014*; *O'Dea et al., 2016*), including Littorinidae (*Reid, Rumbak & Thomas, 1996*; *Williams, Reid & Littlewood, 2003*; *Williams & Reid, 2004*; *Williams & Duda, 2008*).

## Multi-population divergence model selection

For all samples for which we collected multi-locus data, we also used IMa3 (*Hey et al., 2018*), to infer the best-fitting multi-population tree topology under a full isolation-with-migration (IM) model. The best-fitting tree topology was estimated using migration priors with exponential distributions (with a mean of $0.1/\mu$). The phylogeny selection step was conducted on the computing cluster as described above until the run converged after which the sampled values were reloaded until 100,000 genealogies had been sampled.
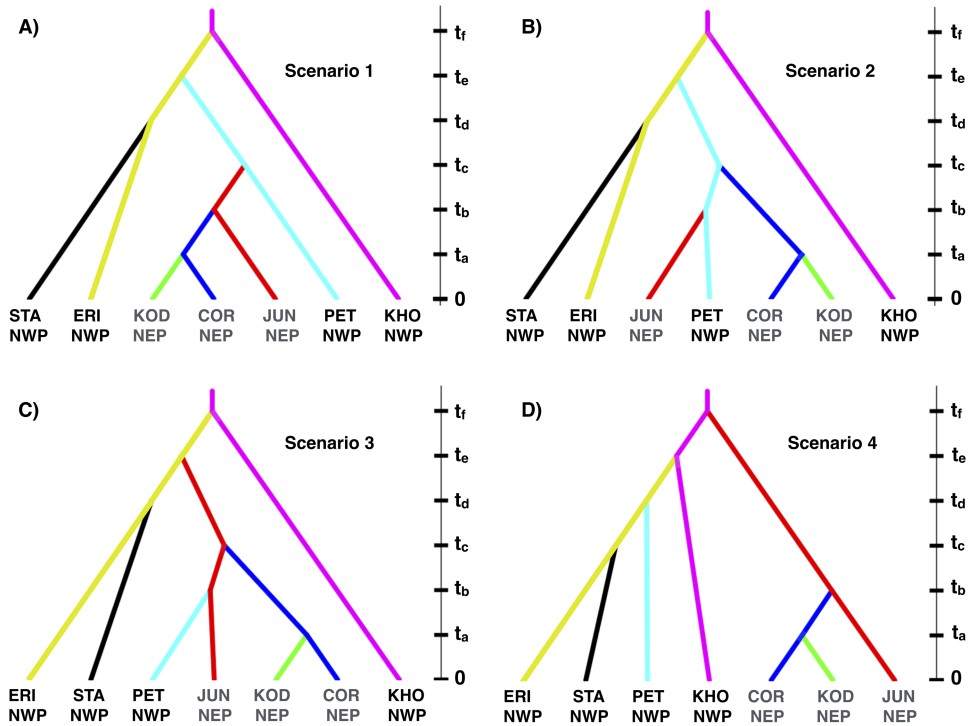

**Figure 1** **Four biogeographic scenarios evaluated with ABC modeling.** Scenarios 1–3 (A–C, respectively) were based on the top (best-fitting) 100 trees from the IMa3 phylogeny selection step, each which contains a monophyletic grouping of northeastern Pacific populations with the Kamchatka population (PET). Scenario 4 (D) represents the null hypothesis that NEP populations share no close relationship to any northwestern Pacific population.

## ABC modeling of biogeographic scenarios

Based on the IMa3 results, we compared four biogeographic scenarios with approximate Bayesian computational (ABC) modeling (Fig. 1). The first three scenarios were based on the top (best-fitting) 100 trees from IMa3, each of which contained a monophyletic grouping of NEP populations with PET in the NWP (see 'Results', below). Scenarios 1 and 2 are consistent with a history in which NEP populations were recently colonized from the NWP (PET) either once (Scenario 1) or twice (Scenario 2). Scenario 3 describes a history in which PET was colonized from a NEP ancestor. Scenario 4 represents the null hypothesis that NEP populations have no close relationship to any NWP population (see *Cox, Zaslavskaya & Marko, 2014*). We evaluated scenarios by analyzing simulated data with the same number of individuals, loci, and sequence lengths as in the empirical data.

Given that we could not detect any evidence of gene flow between any populations with either IMa2 or IMa3 (see 'Results', below), we used DIYABC 2.0.4 (*Cornuet, Ravigne & Estoup, 2010*) to model the biogeographic scenarios. We chose large and broadly overlapping uniform priors (Table S1) for demographic parameters (population sizes and divergence times) based on the pairwise upper and lower 95% highest posterior densities from the IM analyses. The models were evaluated by simulating $1 \times 10^6$ data sets for each of the four scenarios followed by a rejection step using all one-sample statistics and

three two-sample summary statistics (the number of segregating sites, the mean of pairwise differences, and $F_{ST}$) available in DIYABC. The most likely scenario was identified with a polychotomous logistic regression (*Cornuet et al., 2008*; *Cornuet, Ravigne & Estoup, 2010*) computed with 50,000 simulated datasets with summary statistics that were most like those generated from the observed data (*Cornuet et al., 2008*; *Cornuet, Ravigne & Estoup, 2010*).

We assessed confidence in the top two models by estimating the type I and type II error rates from 500 a posteriori simulations of each scenario (*Cornuet, Ravigne & Estoup, 2010*; *Robert et al., 2011*). The false-positive rate for a biogeographic scenario is the proportion of times that a scenario was chosen when the data were simulated under an alternative scenario; the false-negative rate is the proportion of times that an alternative scenario was chosen when the data were simulated under the focal scenario.

## Multi-population parameter estimates

We estimated population parameters using both IMa3 and ABC. For IMa3, we first estimated the priors for the demographic parameters using the best-fitting population tree inferred with IMa3 and DIYABC, with hyperpriors for all demographic parameters, such that the parameter priors are included in the state space and updated over the course of the run. The updated parameter priors from this first step are then used in a second IMa3 step in which population divergence times, effective population size, and gene flow are estimated. Convergence was assessed as described above and parameter estimates were obtained after 1,000,000 genealogies had been sampled. Effective population size and population divergence times were estimated with ABC using a local linear regression on the 1% closest simulated datasets after use of a logit transformation to parameter values (*Cornuet et al., 2008*).

## RESULTS

### Sequence diversity

The most common mtDNA haplotype from populations of *Littorina sitkana* (corresponding to haplotype U9 in *Azuma et al. (2017)*) dominated NEP samples: 83 of 87 individuals (95%) in the NEP had this haplotype, with five of seven NEP samples (spanning more than 2300 km of coastline) fixed for that haplotype (Fig. 2). Only three other haplotypes were found in the NEP, each differing from the most common haplotype at a single nucleotide position. The dominant NEP mtDNA haplotype was also most common in two NWP samples from PET on the Kamchatka Peninsula and STA on the eastern side of Sakhalin Island. Like *Azuma et al. (2017)* we found that most individuals on the Pacific coast of Hokkaido (ERI & NEM) had haplotype K2, a majority of individuals from the third Hokkaido sample on the Sea of Okhotsk (UTO) had haplotypes U2 and U16, and haplotypes U36, U16, and V01 were the most abundant in the Sea of Japan (VOS and KHO).

Although haplotype diversity was variable among samples in both regions, on average, western samples had greater mtDNA haplotype and sequence diversity than eastern samples (Table S2). Only two samples had significant values for any tests of neutrality, but none of these statistics can be calculated when haplotype diversity is zero, as was the case for
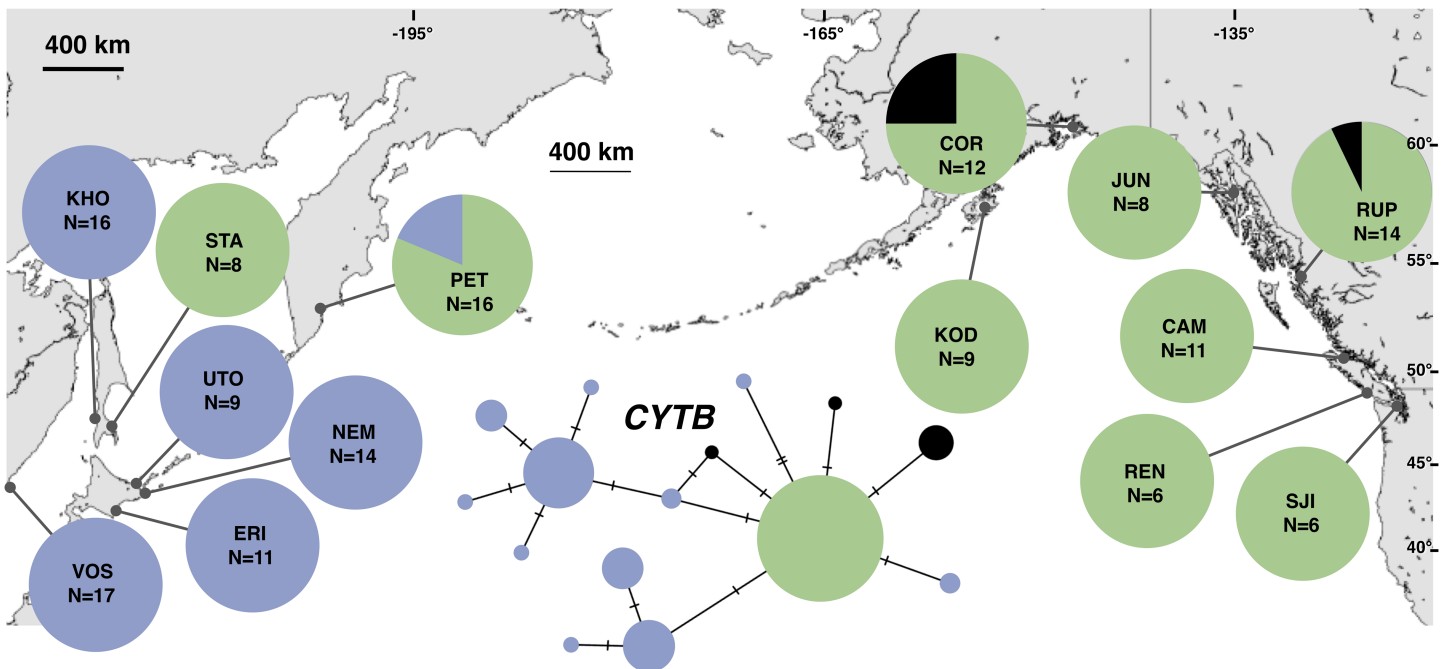

**Figure 2** **Unrooted minimum spanning mitochondrial *CYTB* haplotype network and haplotype frequencies.** Each slash across a branch represents a single mutation. The size of each circle in each network is proportional to the sequence sample size (*N*), with the smallest circle representing one haplotype copy. Blue haplotypes are restricted to the NWP, black restricted to the NEP, and green shared between the two regions.

six out of 14 samples, including most (5 of 7) samples from the NEP (Table S2). All three nDNA genes showed a phylogeographic pattern similar to the mtDNA: NEP samples were dominated by one allele and had lower haplotype and sequence diversity compared to the NWP (Figs. 3–5, Tables S3–S5). As with the mtDNA, the dominant nuclear allele in the NEP was also present in the west, and was always found at PET. As with the mtDNA, few neutrality test statistics were significant for the nuclear loci (Tables S3–S5).

## Spatial patterns of genetic differentiation

Pairwise estimates of $\Phi_{ST}$ for *L. sitkana* show strong and statistically significant mtDNA differentiation between nearly all NWP samples, but no significant differentiation among any NEP samples (Table S6). The majority of pairwise mtDNA $\Phi_{ST}$ estimates between NEP and NWP samples were also large and statistically significant, but some east–west comparisons involving STA and PET yielded relatively low and statistically insignificant values ($\Phi_{ST} \leq 0.110$). All three nuclear loci showed spatial patterns of differentiation that were like those from the mtDNA, with relatively low and non-significant estimates of $\Phi_{ST}$ in the NEP, higher and more statistically significant estimates in the NWP, and similarly high but variable values between NEP and NWP samples, some of which were not statistically significant (Tables S7–S9).
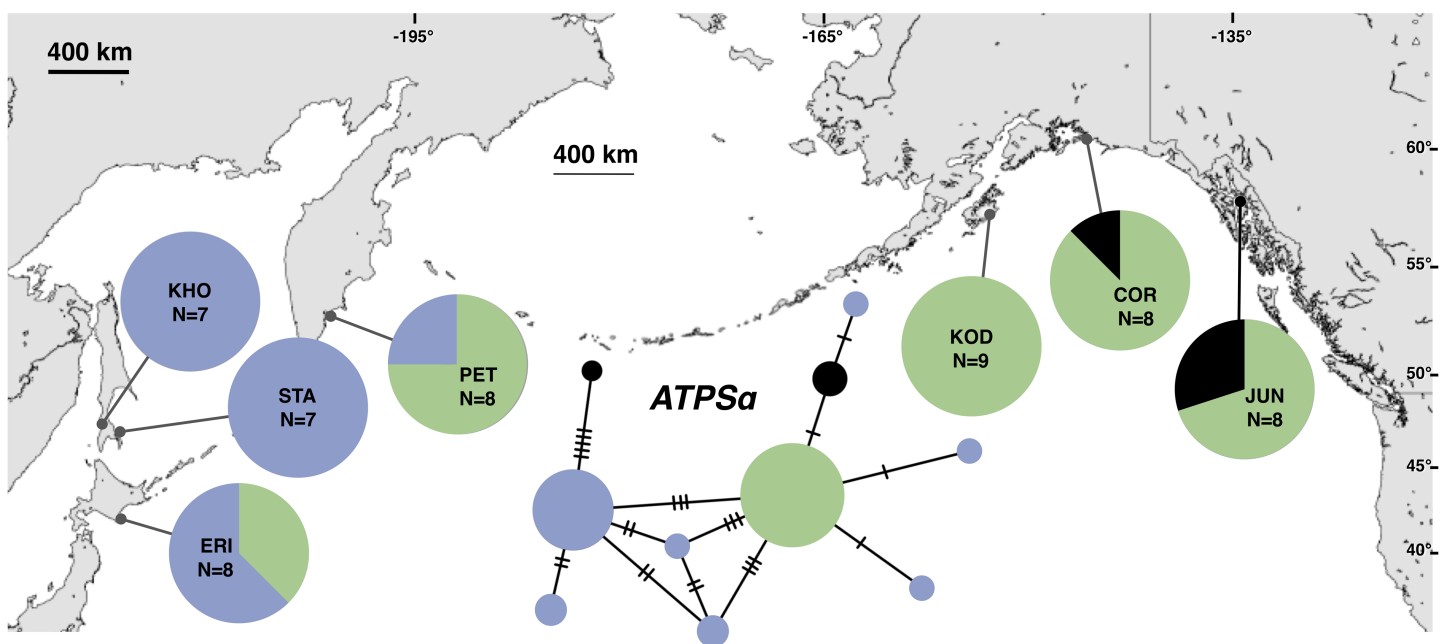

**Figure 3** **Unrooted minimum spanning nuclear *ATPSα* haplotype network and haplotype frequencies.** Each slash across a branch represents a single mutation. The size of each circle in each network is proportional to the sequence sample size (*N*), with the smallest circle representing one haplotype copy. Blue haplotypes are restricted to the NWP, black restricted to the NEP, and green shared between the two regions.

## Pairwise population divergence models

All posterior distributions for the pairwise estimates of the population migration rates (*2 Nm*) between NEP and NWP populations rose asymptotically as *2 Nm* approached zero (not shown), meaning that none were significantly different from zero (Table 2). Many estimates of effective population size in the NEP were relatively small, more than an order of magnitude smaller than estimates for NWP populations (Table 2). Population divergence times (Fig. 6) within the NEP were also uniformly small (≤3,038 years), all with posteriors that overlapped with zero and with a maximum 95% highest posterior density (HPD) of ~74,200. In contrast, population divergence times within the NWP were much larger, ranging from ~107,400 to ~179,900 years, all with posteriors that did not overlap with zero and all with lowest 95% HPDs >19,000 years. Divergence times between NEP and NWP populations were more variable, but several were smaller than most divergence times within the NWP (Fig. 6, Table 2). In one east–west comparison (PET-JUN), the posterior distribution for the population separation time overlapped with zero.

## Multi-population divergence models

In total, 14,219 different tree topologies were sampled by IMa3. Although four trees had the largest product of posterior clade probability (PPCP), no single tree had relatively high support. However, the top 100 trees sorted by PPCP all included a monophyletic grouping of the three NEP populations (COR, JUN, & KOD) and the NWP population from the Kamchatka Peninsula (PET). Among these top 100 trees (Table S10), 62 included

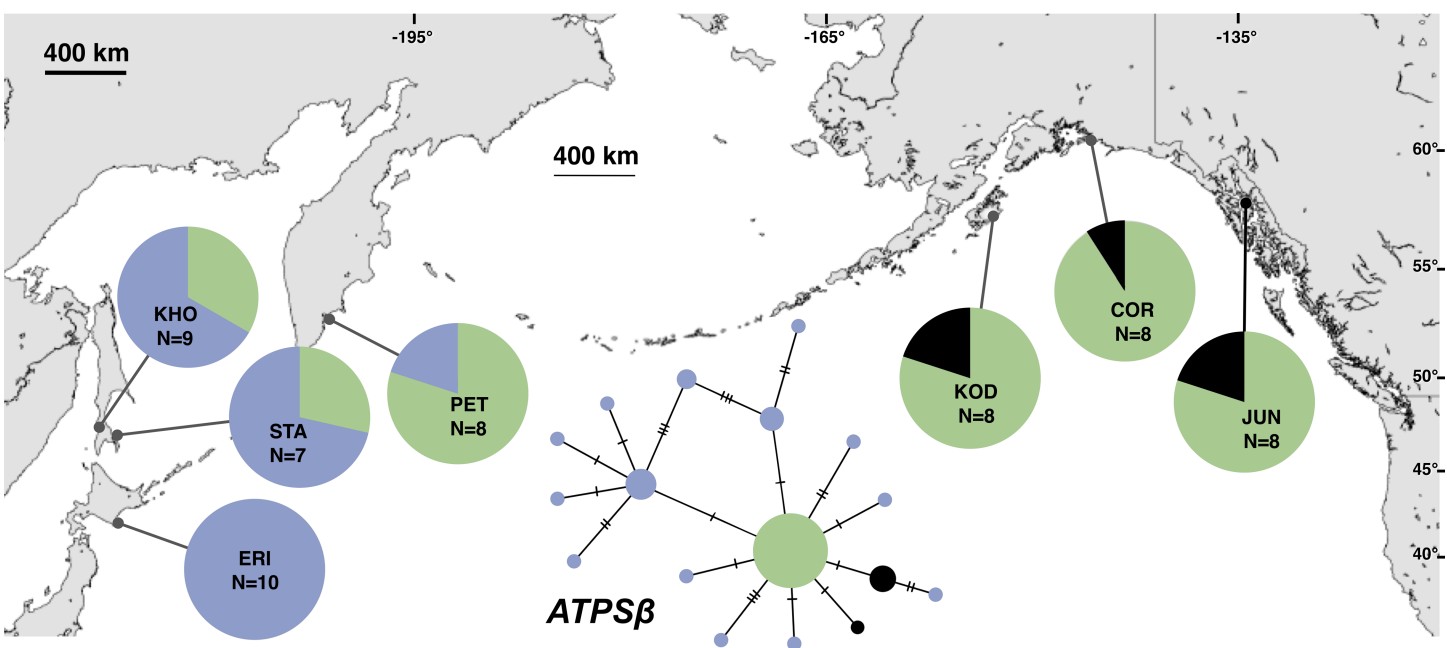

**Figure 4 Unrooted minimum spanning nuclear *ATPSβ* haplotype network and haplotype frequencies.** Each slash across a branch represents a single mutation. The size of each circle in each network is proportional to the sequence sample size (*N*), with the smallest circle representing one haplotype copy. Blue haplotypes are restricted to the NWP, black restricted to the NEP, and green shared between the two regions.

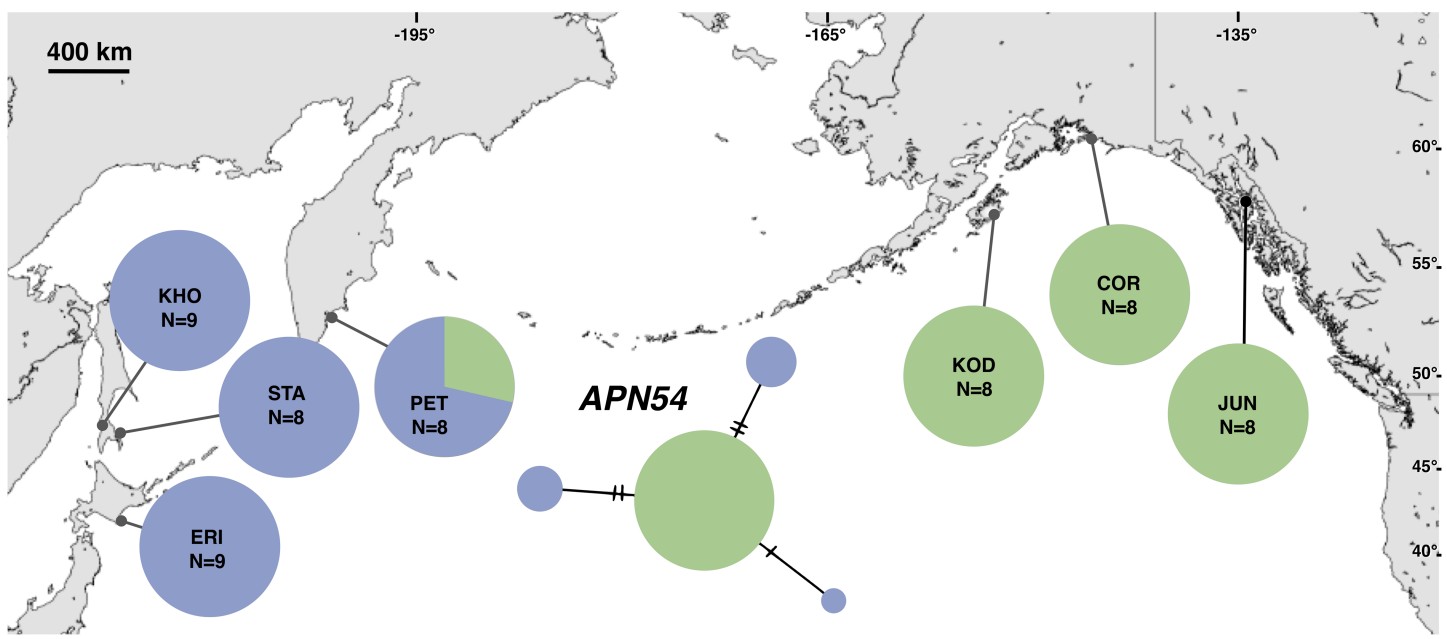

**Figure 5 Unrooted minimum spanning nuclear *APN54* haplotype network and haplotype frequencies.** Each slash across a branch represents a single mutation. The size of each circle in each network is proportional to the sequence sample size (*N*), with the smallest circle representing one haplotype copy. Blue haplotypes are restricted to the NWP, black restricted to the NEP, and green shared between the two regions.

**Table 2  Pairwise population divergence modeling.** Pairwise IMa2 population estimates of current ($N_1$ and $N_2$) and ancestral ($N_A$) effective population size, population migration rates ($2N_1m_{1>2}$ and $2N_2m_{2>1}$) moving forward in time, and divergence time in years ($T$, with highest posterior density or HPD) for all multilocus samples.

| Comparison | $N_1$ | $N_2$ | $N_A$ | $2N_1m_{1>2}$ | $2N_2m_{2>1}$ | $T$ | 95% HPD |
|---|---|---|---|---|---|---|---|
| NWP vs NEP | | | | | | | |
| PET-KOD | 164,538 | 8,829 | 121,197 | 0 | 0 | 35,316 | 1,926–221,953 |
| STA-KOD | 49,348 | 1,013 | 7,091 | 0 | 0 | 18,909 | 1,466–61,668 |
| KHO-KOD | 150,851 | 8,510 | 155,492 | 0 | 0 | 96,957 | 11,965–425,374 |
| ERI-KOD | 297,311 | 8,426 | 164,905 | 0 | 0 | 70,215 | 6,179–373,544 |
| PET-COR | 156,922 | 88,992 | 190,852 | 0 | 0 | 73,786 | 3,629–240,710 |
| STA-COR | 162,007 | 33,365 | 45,059 | 0 | 0 | 77,965 | 21,463–176,660 |
| KHO-COR | 209,098 | 73,960 | 265,709 | 0 | 0 | 189,436 | 47,237–466,041 |
| ERI-COR | 327,359 | 72,359 | 305,290 | 0 | 0 | 123,015 | 28,036–470,891 |
| PET-JUN | 137,896 | 16,546 | 118,984 | 0 | 0 | 31,309 | 0–155,058 |
| STA-JUN | 49,676 | 1,916 | 3,685 | 0 | 0 | 28,578 | 2,820–60,466 |
| KHO-JUN | 154,453 | 16,836 | 132,493 | 0 | 0 | 122,196 | 24,595–447,010 |
| ERI-JUN | 310,940 | 19,363 | 110,480 | 0 | 0 | 87,701 | 30,297–512,917 |
| NEP vs NEP | | | | | | | |
| KOD-COR | 7,287 | 39,690 | 59,276 | 0 | 0 | 1,565 | 0–69,402 |
| KOD-JUN | 2,030 | 9,699 | 1,386 | 0 | 0 | 646 | 0–22,631 |
| COR-JUN | 24,929 | 15,018 | 57,067 | 0 | 0 | 3,038 | 0–74,219 |
| NWP vs NWP | | | | | | | |
| ERI-PET | 302,549 | 101,329 | 168,881 | 0 | 0 | 96,969 | 19,547–336,133 |
| ERI-STA | 214,996 | 201,823 | 51,279 | 0 | 0 | 179,900 | 79,537–327,935 |
| ERI-KHO | 235,872 | 135,648 | 171,936 | 0 | 0 | 146,073 | 42,854–488,908 |
| KHO-PET | 153,504 | 110,548 | 139,607 | 0 | 0 | 149,924 | 43,461–485,485 |
| KHO-STA | 108,564 | 241,188 | 60,444 | 0 | 0 | 107,351 | 38,496–220,649 |
| PET-STA | 84,666 | 145,322 | 62,762 | 0 | 0 | 110,305 | 29,879–302,608 |

**Notes.**

NWP, Northwest Pacific; NEP, Northeast Pacific.

a monophyletic clade of NEP populations (Trees 1–19, 38–41, 62–100); the remaining 38 (Trees 20–37, 42–61) placed PET as the sister-population to JUN.

### ABC modeling of biogeographic scenarios

Scenario 1 received significantly greater support (posterior probability = 0.821, 95% credible interval: 0.785–0.857) over Scenario 3 (posterior probability = 0.099, 95% credible interval: 0.075–0.123), the next best-fitting population history (Fig. 7). *A posteriori* simulations of Scenario 1 showed that the type I error rate was only 15.3%. Based on simulations of the other three scenarios, the average type II error rate was 16.9%.

### Multi-population parameter estimates

Based on the outcome of the topology selection steps, we used Tree 1 (Scenario 1) to estimate demographic parameters with IMa3 and DIYABC. Going backwards in time, IMa3 inferred that NEP populations coalesced down into a common ancestor ~21,400 years ago ($t_1$, Fig. 8), merging with PET in the NWP ~46,900 years ago ($t_2$, Fig. 8); the latter population
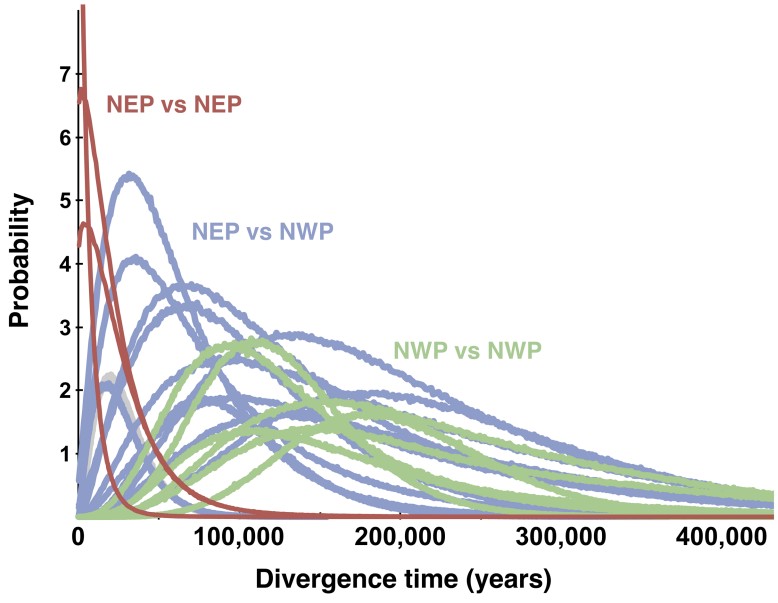

**Figure 6  Multilocus posterior probability distributions for pairwise population divergence times.**
Each curve is from a single pairwise comparison among samples in the northeastern Pacific (NEP) and
northwestern Pacific (NWP).

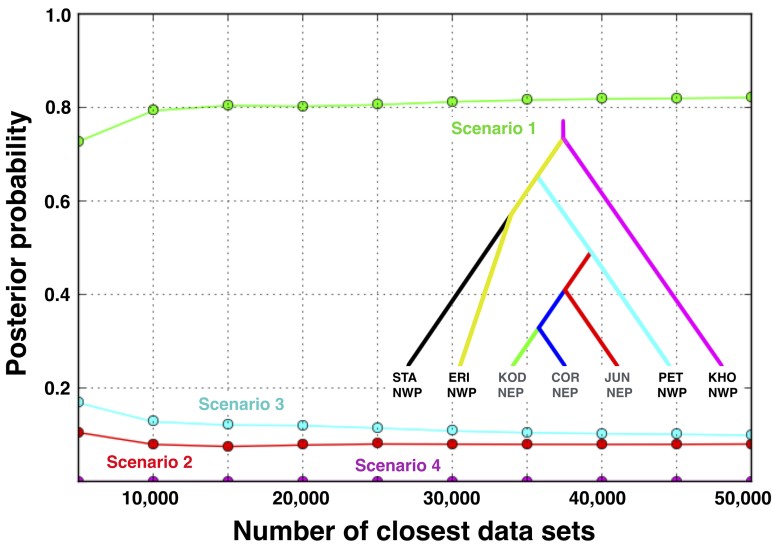

**Figure 7  Posterior probabilities for four biogeographic scenarios evaluated with ABC modeling.** The
most likely scenario (Scenario 1) was identified with a polychotomous logistic regression computed with
50,000 simulated datasets with summary statistics that were most like those generated from the observed
data.

fusion coincided with a quadrupling in effective population size (Table 3). The posterior
probabilities for all 42 population migration rate ($2Nm$) parameters either overlapped with
zero or rose asymptotically as $2Nm$ approached zero (not shown), indicating that no gene

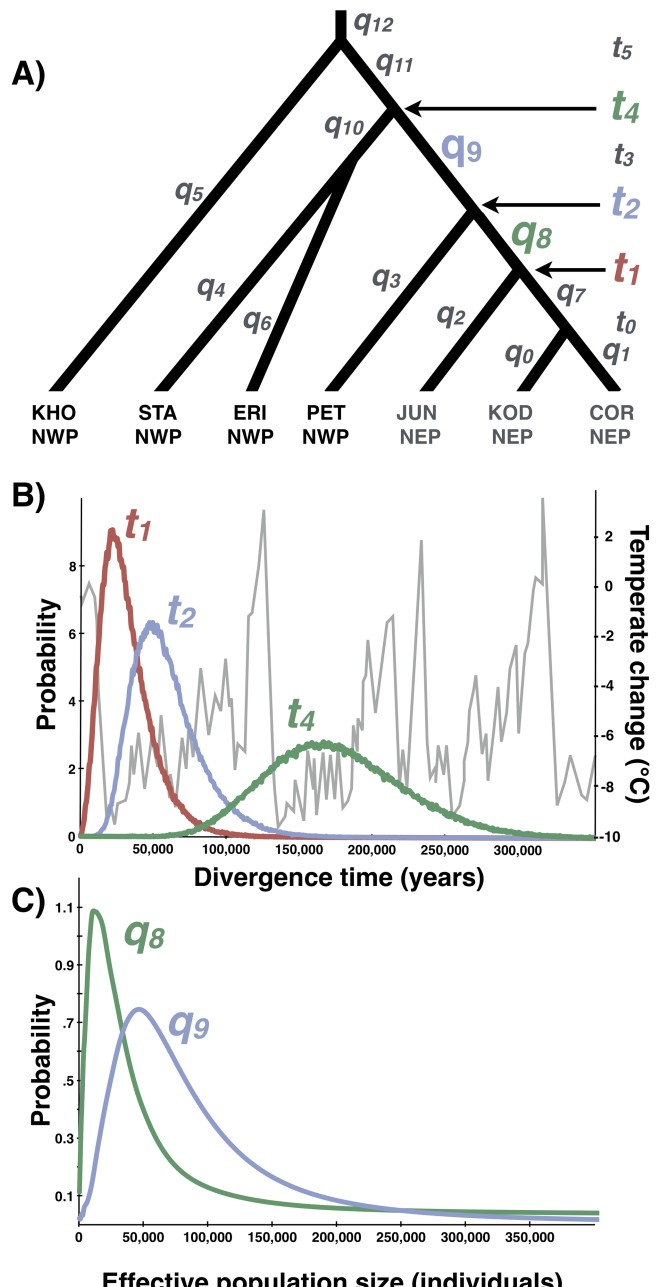

**Figure 8** **Multilocus demographic parameter estimates from IMa3.** (A) Best-fitting tree topology. Labels on branches and nodes correspond to effective population sizes ($q_{0-12}$) and divergence times ($t_{0-5}$) in Table 3. (B) Posterior probability distributions for divergence times between northeastern Pacific (NEP) and northwestern Pacific (NWP) populations and corresponding temperature relative to present (data from *Petit et al. (1999)*). (C) Posterior probability distributions for ancestral population sizes ($q_{8-9}$).

flow could be detected. The parameter estimates from DIYABC were similar, indicating that the NEP populations coalesced down to a common ancestor ~22,400 years ago and merged with PET ~54,400 years ago (Table 4).

**Table 3 Demographic parameter estimates from the multi-population analysis with IMa3.** Most probable estimate (MPE) and 95% confidence intervals for effective population size ($q_{0-12}$, individuals) and divergence time ($t_{0-5}$, years). The posterior probabilities for all 42 different population migration rate ($2Nm$) parameters either rose asymptotically as 2Nm approached zero or broadly overlapped with and are therefore not shown.

| Parameter | Description | MPE | 95% CI |
|---|---|---|---|
| $q_0$ | Population size: KOD | 8,448 | 0–820,361 |
| $q_1$ | Population size: COR | 280,391 | 98,063–985,063 |
| $q_2$ | Population size: JUN | 22,380 | 4,319–702,427 |
| $q_3$ | Population size: PET | 80,966 | 20,750–565,946 |
| $q_4$ | Population size: STA | 209,734 | 73,012–741,743 |
| $q_5$ | Population size: KHO | 92,821 | 31,458–268,365 |
| $q_6$ | Population size: ERI | 242,740 | 106,389–628,344 |
| $q_7$ | Ancestral population size: KOD & COR | 9,714 | 0–779,262 |
| $q_8$ | Ancestral population size: KOD, COR, & JUN | 11,258 | 0–664,233 |
| $q_9$ | Ancestral population size: KOD, COR, JUN, & PET | 46,603 | 1,503–334,038 |
| $q_{10}$ | Ancestral population size: STA & KHO | 211,003 | 49,613–115,848 |
| $q_{11}$ | Ancestral population size: KOD, COR, JUN, PET, STA, & KHO | 760,157 | 90,653–883,976 |
| $q_{12}$ | Ancestral population size: KOD, COR, JUN, PET, STA, KHO, & ERI | 78,346 | 17,149–233,693 |
| $t_0$ | Splitting time: KOD & COR | 7,970 | 0–36,661 |
| $t_1$ | Splitting time: NEP (KOD, COR, & JUN) | 21,359 | 4,144–67,266 |
| $t_2$ | Splitting time: NEP (KOD, COR, & JUN) from NWP (PET) | 46,863 | 19,447–107,434 |
| $t_3$ | Splitting time: STA from KHO | 102,971 | 45,588–189,683 |
| $t_4$ | Splitting time: STA & KHO from PET, KOD, COR, & JUN | 166,092 | 87,669–269,382 |
| $t_5$ | Splitting time: ERI from STA, KHO, PET, KOD, COR, & JUN | 214,549 | 120,823–321,665 |

**Table 4 Demographic parameter estimates from ABC modeling.** Posterior modal estimates (95% credible interval) of parameter values from the most probable biogeographic history (Scenario 1) inferred using ABC modeling with DIYABC. Single parameter estimates are in units of effective population size ($n_{1-7}$) or time ($t_{a-f}$) in generations.

| Parameter | Description | Mode | 95% CI |
|---|---|---|---|
| $n_1$ | COR $N_e$ | 144,000 | 53,000–197,000 |
| $n_2$ | KOD $N_e$ | 19,900 | 5,450–177,000 |
| $n_3$ | JUN $N_e$ | 21,200 | 7,730–158,000 |
| $n_4$ | PET $N_e$ | 99,800 | 33,700–420,000 |
| $n_5$ | ERI $N_e$ | 321,000 | 128,000–458,000 |
| $n_6$ | KHO $N_e$ | 88,800 | 34,600–377,000 |
| $n_7$ | STA $N_e$ | 315,000 | 122,000–485,000 |
| $t_a$ | Splitting time: KOD from COR | 2,810 | 687–44,500 |
| $t_b$ | Splitting time: COR from JUN | 22,400 | 7,730–120,000 |
| $t_c$ | Splitting time: JUN from PET | 54,400 | 26,400–223,000 |
| $t_d$ | Splitting time: STA from ERI | 205,000 | 98,100–363,000 |
| $t_e$ | Splitting time: PET from ERI | 244,000 | 111,000–441,000 |
| $t_f$ | Splitting time: KHO from ERI | 483,000 | 258,000–592,000 |

## DISCUSSION

Our study provides strong support for *Azuma et al.*'s (*2017*) hypothesis that *Littorina sitkana* recently colonized the northeastern Pacific (NEP) from the northwestern Pacific (NWP). The primary evidence for this conclusion is the extremely recent divergence times among all populations within the NEP and between all NEP populations and some populations in the NWP on Sakhalin Island and the Kamchatka Peninsula. This recent biogeographic connection is evident in the haplotype network of each locus, as the most common allele in the NEP is always the most abundant allele at three of four loci in our new sample from Kamchatka, the northeastern most sampled population in the NWP. Much smaller effective population sizes in the NEP is also consistent with the recent colonization hypothesis, as is the large reduction in population size in the ancestral NEP population (i.e., $q9 \rightarrow q8$, Fig. 8).

Consistent with previous studies of *L. sitkana* (*Kyle & Boulding, 2000*; *Lee & Boulding, 2009*; *Marko et al., 2010*; *Azuma et al., 2017*), we found very little genetic diversity and no significant population subdivision at all four loci in the NEP, an unexpected pattern for a species that lacks planktonic larvae, but one that is readily explained by a recent range expansion. The consistency of the pattern across loci allows us to rule out the possibility of a mtDNA selective sweep in the NEP, or other hypotheses about natural selection acting directly or indirectly on the genetic markers themselves. Instead of a colonization event, a severe bottleneck on NEP population size during the LGM might cause the loss of most haplotypes (but retention of the most common haplotype) across many loci, creating a pattern similar to a founder effect. However, the most important piece of evidence in this study, evident in both the IMa3 and DIYABC analyses, is that the Kamchatka population (PET) shares a more recent common ancestry with NEP populations than the Kamchatka population does with any other NWP population, a result that can only be explained by the hypothesis that *L. sitkana* recently dispersed to North America from the NWP.

Lacking a planktonic life history stage, colonization of the NEP was presumably accomplished by rafting of adults or egg masses on floating biological debris (*Knox, 1960*; *Johannesson, 1988*; *O'Foighil et al., 1999*; *Collin, 2001*; *Colson & Hughes, 2004*; *Waters & Roy, 2004*; *Thiel & Haye, 2006*; *Gordillo & Nielsen, 2013*; *Cumming et al., 2014*). Although both the ABC and full-likelihood methods yielded similar results, the arrival of *L. sitkana* in North America is probably best estimated by the full-likelihood methods (*Hickerson, Dolman & Moritz, 2006*). The multi-population IMa3 method conservatively bounded this event between 107,400 and 4,100 years ago, the upper bound on the separation of NEP populations from PET ($t_2$, Fig. 8B) and the lower bound on the time at which NEP populations coalesced down into a common ancestor, respectively ($t_1$, Fig. 8B, also see Table 3). The four most recent pairwise IMa2 population estimates of divergence times between NWP and NEP populations (Table 2) all broadly overlapped with this window.

The conventional biogeographic wisdom used to explain how species move between the NEP and NWP is that warm interglacial climates permit stepping-stone colonization across the Aleutian-Commander Archipelago (ACA), whereas cold glacial climates result in extinction of northern populations and the closure of this dispersal corridor (*Vermeij, 1989*;
*Cox, Zaslavskaya & Marko, 2014*; *Azuma et al., 2017*). Although the confidence interval on the dispersal window inferred from the multi-population method extends back through nearly all of the last glacial period (to 107,400 years ago), the idea that *L. sitkana* colonized the NEP as the climate warmed since the LGM seems likely given the peak in the highest posterior densities for both the divergence time between NEP populations and PET ($t_2$, Fig. 8B) and the divergence time among NEP populations ($t_1$, Fig. 8B) are both closer to the end of the LGM than the peak of the last interglacial 125,000 years ago; the arrival time of *L. sitkana* in the NEP is also probably closer to the separation times among NEP populations rather than the separation time between the ancestral population that split away from PET in the NWP. Slightly older than expected divergence times from the molecular data might be a consequence of the potential time-dependency of molecular rates (*Ho et al., 2011*). Because the substitution rate that we used was based on fossil calibrations, it could be an underestimate of the instantaneous mutation rate (*Ho et al., 2007*; but see *Woodhams, 2006*; *Emerson, 2007*; *Navascues & Emerson, 2009*), a more appropriate rate for intraspecific phylogeography (*Crandall et al., 2012*). Using a faster rate in our analyses, however, would only strengthen the idea that *L. sitkana* recently colonized the NEP. That said, a gross mismatch between the substitution rate and the instantaneous mutation rate seems unlikely given that the point estimates for divergence times within the NEP are <3,100 years and the 95% HPDs overlap with zero. This cross-validation suggests that our fossil-based rates and divergence times estimates are not heavily biased by time-dependency.

Dispersal across the ACA is probably frequent given the large number of rocky shore species whose ranges span the entire archipelago (*Vermeij, Palmer & Lindberg, 1990*), although this inference assumes that most species were driven out of the ACA during cold glacial climates. Among molluscs whose range endpoints are found within the ACA, most are NEP species, outnumbering NWP species by more than four to one, suggesting that species may have been more likely to travel from east to west than west to east, presumably in the predominantly westward flow of the Alaska Current (*Vermeij, Palmer & Lindberg, 1990*). Alternatively, the range endpoint data might indicate that biogeographic barriers in the ACA are stronger for NEP species. Despite the large number of taxa distributed across the entire archipelago, *L. sitkana* is the only species studied with genetic data that shows a phylogeographic history consistent with post-glacial dispersal in either direction between the NWP and NEP. Several other species with amphi-Pacific distributions show large differences in genetic diversity between NWP and NEP populations, but all show relatively high levels of genetic differentiation between NWP and NEP populations (*Sato et al., 2004*; *Cassone & Boulding, 2006*; *Liu et al., 2007*; *Canino et al., 2010*), including reciprocal monophyly of mtDNA lineages (e.g., *Cox, Zaslavskaya & Marko, 2014*) that indicates an extended period of isolation with no gene flow. Strong east–west differentiation in these taxa suggests that some of the single nominal species with ranges that span the ACA may be recently-diverged sister species that have come into secondary contact in the ACA, rather than "eastern" or "western" species that have recently dispersed from one end of the ACA to the other.

The relatively sparse fossil record provides little complementary information about changes in rocky shore species distributions, yielding a few snapshots of communities captured in uplifted sedimentary rocks formed during past sea level high-stands (e.g., *Valentine & Jablonski, 1993*). However, each of the congeners *L. scutulata*, *L. plena*, and *L. keenae* all are relatively abundant in the NEP Pleistocene fossil record (e.g., *Lindberg & Lipps, 1996*) and all have much greater mtDNA diversity in the NEP than *L. sitkana* (*Kyle & Boulding, 2000*; *Lee & Boulding, 2007*; *Lee & Boulding, 2009*; *Marko et al., 2010*), together indicating that these three species have been relatively abundant in the NEP throughout the Pleistocene. In contrast, only a single record of a junior synonym of *L. sitkana* (*L. sitkana* var. *atkana* Dall, 1886) has been reported from the NEP Pleistocene fossil record (*Zullo, 1969*), from a marine terrace in southern Oregon dated at 80,000 years old (+/−5,000 years), deposited during the last glacial, but during a brief period in which summer insolation briefly rose in the northern hemisphere (*Muhs et al., 2006*). Although falling within the broad window (107,000–4,140 years ago) inferred for the arrival of *L. sitkana* in North America, the age of the fossils falls outside the upper bound on the separation times among populations further north in Alaska (∼67,300 years, Table 3), populations that should be older if *L. sitkana* dispersed across the ACA, arriving first in Alaska.

However, we and two other taxonomic authorities (DG Reid & AV Chernyshov) inspected photographs of the fossils (UCMP B-7493) and were unable to provide a positive identification, as the relatively small and smooth shells could be either *L. sitkana* or *L. subrotundata* (for a morphological comparison see *Reid, 1996*). Compared to *L. sitkana*, NEP populations of *L. subrotundata* have four times as many mtDNA haplotypes and greater genetic differentiation among populations in British Columbia (*Kyle & Boulding, 2000*), suggesting an older history in the NEP and that the 80,000-year-old fossils are *L. subrotundata*. If the fossils are actually *L. sitkana*, a first occurrence 80,000 years ago at a relatively low latitude is possible if *L. sitkana* first arrived in the NEP by dispersing directly across the Pacific in the North Pacific Current, a plausible dispersal route highlighted by the recent transport of NWP taxa to the NEP on tsunami debris (*Carlton et al., 2017*). The genealogical connection between Kamchatka and NEP populations of *L. sitkana* discovered here is not consistent with this idea, unless colonists from Kamchatka were carried south in the Oyashio Current, later merging with the Kuroshio Current, and eventually moving eastward in the North Pacific Current. Genetic analysis of samples from the Aleutian Islands can resolve this question, as stepping-stone colonization across the ACA is expected to create a cline in allele frequencies and gradual decline in genetic diversity (*Slatkin & Excoffier, 2012*) from west to east.

## CONCLUSIONS

In summary, analysis of multi-locus sequence data indicates that extremely low genetic diversity among NEP populations of *L. sitkana* is best explained by a recent and rapid range expansion from the NWP. Although the founder events associated with the spread of *L. sitkana* throughout the NEP depleted mtDNA and nuclear DNA diversity, the four

loci that we used contained enough information to estimate the arrival of *L. sitkana* in North America between 107,400 and 4,100 years ago. Reductions in genetic diversity may ultimately limit range expansions (e.g., *Travis et al., 2007*; *Frankham, 2010*; *Hallatschek & Nelson, 2010*; *Polechová, 2018*), but the loss of diversity in NEP populations of *L. sitkana* did not appear to hamper its expansion throughout the Pacific coast of North America. Further southward expansion may be limited instead by other factors, such as habitat availability, predation, and desiccation of egg masses (*Behrens Yamada, 1977*; *Behrens Yamada, 1976*; *Behrens Yamada, 1992*). Competition could also play a role, as the habitat of *L. sitkana* shifts from predominantly wave exposed shores in the NWP (*Golikov & Kusakin, 1962*; Golikov & Kusakin, 1978; *Golikov & Scarlato, 1967*) to primarily sheltered shores in the NEP, particularly in British Columbia, Washington, and Oregon (*Harger, 1972*; *Behrens, 1972*; *Boulding & Van Alstyne, 1993*) where *L. sitkana* appears to be replaced by *L. subrotundata* on exposed shores (*Boulding, 1990*; *Boulding & Van Alstyne, 1993*). Similar range-wide analyses of genetic diversity in other amphi-Pacific taxa, particularly *L. subrotundata*, will be informative, as they can provide insight into the proportion of NEP and NWP that are recent colonists and how depletion of genetic diversity during post-glacial colonization events may impact future adaptation and range shifts in response to climate change.

## ACKNOWLEDGEMENTS

We thank M Yanagida and K Vicknair for logistical support and V Brykov for additional financial support in Russia. V Duriev provided transportation on Sakhalin Island. The manuscript was improved with comments in reviews from K Johannesson and E Crandall.

### Funding

This work was supported by the National Science Foundation (NSF OCE-0550526 to Peter Marko) and received additional financial support in Russia from V Brykov. The funders had no role in study design, data collection and analysis, decision to publish, or preparation of the manuscript.

### Grant Disclosures

The following grant information was disclosed by the authors:
National Science Foundation: NSF OCE-0550526 to Peter Marko.

### Competing Interests

The authors declare there are no competing interests.

### Author Contributions

- Peter B. Marko conceived and designed the experiments, performed the experiments, analyzed the data, contributed reagents/materials/analysis tools, prepared figures and/or tables, authored or reviewed drafts of the paper, approved the final draft.
- Nadezhda I. Zaslavskaya conceived and designed the experiments, performed the experiments, approved the final draft, edited manuscript.

## Field Study Permissions

The following information was supplied relating to field study approvals (i.e., approving body and any reference numbers):

Collections in Alaska were obtained with permission from the Alaska Department of Fish and Game. In Japan, collections were approved in person at local offices of the Hokkaido Federation of Fisheries Cooperative Associations in Erimo, Nemuro, and Utoro. Collections of marine invertebrates in Russia required no permits.

## DNA Deposition

The following information was supplied regarding the deposition of DNA sequences:

New nuclear and mitochondrial DNA sequences are available at GenBank: MN120839–MN120881.

## Data Availability

The raw data are available at GenBank: MN120839–MN120881.

## Supplemental Information

Supplemental information for this article can be found online at http://dx.doi.org/10.7717/peerj.7987#supplemental-information.

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
