# Peer review of "Geographic origin and timing of colonization of the Pacific Coast of North America by the rocky shore gastropod Littorina sitkana"

_PeerJ, doi:10.7717/peerj.7987_

## Round 0.1 · original submission · Minor Revisions

Both reviewers were complimentary about your paper, and I agree with their views. The only major issue made is by Reviewer 2, who asks for some more specific information on mutation rate used. This will have a bearing on time estimates. Please clarify this point. Other than that, the reviewers make a number of helpful suggestions to improve presentation. I recommend use of italics for all parameters- currently only some are italicised. Please also pay attention to the Reference section- I see some journal titles un-italicised, or partly so.

·

Basic reporting

No comment.

Experimental design

No major comment.
I have made some minor comments listed among general comments below.

Validity of the findings

No comment.

Additional comments

This is a neat study showing clear evidence for a recent expansion and colonization of the eastern coast of North Pacific by a western North Pacific rocky shore snail. The problem is well presented and the data consistently supports the conclusion. I think it is also a nice contribution since it highlights the problem of understanding population genetic patterns without considering the demographic history of a species. In this particular case, it was for a long time confusing that despite poor dispersal (in the absence of a planktonic larval stage) this species did not show large differentiation among populations.

I have only minor comments that the authors might like to consider to improve clarity of argumentation.

Title: Maybe add "the Pacific coast of North America" to be a bit more precise.

Abstract. First sentence if a bit technical. I would suggest raising already here the point that the demographic history of a species may play a fundamental role in shaping contemporary population genetic structure.
The same argument may go also for the Introduction, where a couple of more conceptually directed sentences would be good before getting into the technical details of mtDNA as a tool for phylogeography.

Line 76. Here we use "additional samples" - it is not really clear what samples already existed. I guess it would be easier to leave these details to the Materials and Methods section. This said, there is of course the need to explain in what way the current manuscript takes us beyond what Azuma et al. 2017 already concluded, so the rest of this paragraph is really crucial.

Line 159. It is not completely clear to me why this 72hours intervals were done. Was it in order to be able to evaluate convergence of starting conditions? Or was it some more practical reason?

Table 1. Can number of snails sequenced also be indicated in this table?

Fig. 1. In Scenario 2, PET and JUN have shifted affiliations (NEP - NWP). (In general, figures are nice and illustrative.)

Lines 228-231 and Fig. 2 and Figs. 3-5. There is no explanation for the colours used to indicate haplotypes but I guess blue is for all haplotypes only found in NWP and black is for those only found in NEP while green are found in both areas. I can see the point in this simplification of the illustration, but at the same time this does not really give the impression of the higher number of haplotypes in the NWP area, and what is completely lost is the information of the distribution of these haplotypes. Is it possible to use different colours for haplotypes and then overlaying, different rasters for the two main areas?

Fig. 5. The pattern for this gene is a bit surprising - that is, that the most common haplotype is no longer present in 3 of 4 NWP sites. Of course, it looks like this also because this haplotype is so strongly represented in NEP. Is the green haplotype the most common one in the PET population?

Line 233. This sentence is a bit contradictory - in particular, as you cannot see from the figures how many haplotypes are present in the different NWP sites.

Line 238 (Table S2). N values are indicated here, but I think these can also be indicated in Table 1. In addition, many of the supplement tables lack a heading or appropriate description. Although much is evident, I think a short description, for example, indicating what bold figures means, should follow with each supplement table.

Line 242. I think there is a word missing somewhere.

Line 250. In Tables S6-S9, boldface letters show comparisons NEP and NWP and not if Fst are significant or not while in earlier tables bold face meant (I think) significance. There is furthermore no description of what stars means (but probably significance). It would be appropriate with Table headings for all tables!

Line 263. Effective population sizes are estimated from models and shown to differ considerably between NEP and NWP populations. Still this result is not brought up in the Discussion.

Fig. 8. Colours in C) do not match colours in A).

Line 318. The role of the PET population is a bit hard to appreciate. Do you here mean that the bottleneck appeared before PET was established? I think I am confused about the "but" in the middle of the sentence. Is not a bottleneck compatible with PET being rather isolated from the rest of the NWP and all NEP derived from an initial migration from PET to NEP?

Line 426. Remove one of the two "for"

·

Basic reporting

Manuscript was tight and well-written. Sufficient context was provided. Results were relevant to hypotheses.

Experimental design

Research question was well defined, and the investigation and analyses were performed rigorously. A key piece of information was missing though: the mutation rate used to convert the model parameters to years and population sizes.

Validity of the findings

Findings will be valid once uncertainty in mutation rate is addressed.

Additional comments

## Summary

This is tight and well-executed study of divergence in an amphi-Pacific periwinkle *Littorina sitkana*. The authors obtain DNA sequence from mitocondrial cytochrome B and 3 nuclear loci, and then use divergence modeling in full likelihood and approximate Bayesian contexts to demonstrate convincingly that the Northeast Pacific populations of this species were colonized from the Northwest Pacific between ~4K and 108K years ago.

Overall I enjoyed reading this paper. I found the methods to be rigorous and the discussion and conclusions to follow nicely from the presented results. I commend the authors on doing some "old-school" Sanger sequencing and making careful and robust inferences based on a handful of loci, rather than struggling with >10K loci of unknown provenance available from RAD-seq. I even picked up some tips that I intend to apply within my own research.

## Major Comments

My only major concern is with the mutation rate used to translate model parameters into units of time and population size. The authors don't present the mutation rate that they used, referring to a 23 year old book of David Reid's as their source and describing it as a fossil calibration. This lack of transparency makes me nervous. The authors are clearly aware of Simon Ho's time-dependency hypothesis, which has a lot of support. Depending on what rate they actually used, the true rate could be 2 to 5 times greater than that, with inferred times of divergence therefore decreasing by the same magnitude. The authors acknowledge this in their discussion, but I'd like to see a more complete discussion of how uncertainty in the mutation rate might impact their inference. I refer them to my own work on time dependency, which suggests that the decrease in apparent rate over time is especially steep in marine invertebrates (Crandall et al. 2012; although there is a typo in the rate slope in that paper; see the corrigendum)

## Specific Comments

L54 Of course a selective sweep is a third possibility for low diversity
L95 Gastropods can be difficult to preserve because they close up their operculum. How were these samples preserved?
L130 What criterion was used within ModelTest?
L167 What was this rate? Please say more about how it was estimated.
L261 It would be helpful to refer to Table 2 and Figure 6 at the beginning of this paragraph
Figures 2-5 - Could the sample size in either individuals or haplotypes be added within or next to each population pie chart?
Figure 8 - The colors for q8 and q9 are reversed in plate a and c
Supplemental Tables - It would be nice if each table of summary statistics was labeled with the marker it is describing. Similarly for PhiST tables. Are these tables showing FST or PhiST? The text suggests PhiST, but the download button says FST

## Literature Cited

Crandall ED, Sbrocco EJ, DeBoer TS, Barber PH, Carpenter KE. 2012. Expansion Dating: Calibrating Molecular Clocks in Marine Species from Expansions onto the Sunda Shelf Following the Last Glacial Maximum. Mol Biol Evol 29:707–719.

---

## Round 0.2 · accepted · Accept

Thank you for your attention to the points raised by the reviewers. I notice that several journal names are still not italicised- please attend to this at the proof stage.